# HICL-TBI: Hyperbolic In-Context Learning for Traumatic Brain Injury Outcome Prediction

## Abstract

Tabular foundation models like TabPFN excel at training-free In-Context Learning (ICL), yet standard retrieval strategies rely on Euclidean geometry, failing to capture the hierarchical progressions inherent in clinical data. In this paper, we introduce HICL-TBI, a novel Hyperbolic ICL framework that integrates Poincaré Ball geometry to resolve this representational bottleneck. To robustly handle real-world data imperfections, we propose Hyperbolic Fréchet Imputation to reconstruct missing values without flat-space distortion, and Density-Aware Hyperbolic Retrieval to dynamically adapt the context size based on local manifold density. By mapping mild cases near the origin and exponentially separating severe outliers, our approach effectively mitigates class imbalance and the Euclidean crowding problem. Extensive experiments on three real-world Traumatic Brain Injury (TBI) datasets demonstrate that HICL-TBI outperforms Euclidean baselines in fine-grained classification and small-data regimes, all while maintaining a strictly training-free pipeline.

## 1. Introduction

Tabular Foundation Models (FMs), such as TabPFN (Hollmann et al., 2023), have initiated a paradigm shift in small-data regimes, enabling robust zero-shot predictions via In-Context Learning (ICL) (Moor et al., 2023). However, similar to Large Language Models, their reasoning capacity is strictly bottlenecked by the context window size. To bypass this, retrieval-augmented strategies (e.g., LoCalPFN (Thomas et al., 2024), TabR (Gorishniy et al., 2024)) have emerged to construct refined local contexts. Yet, they predominantly rely on Euclidean ($L_2$) distance, implicitly assuming a flat data manifold and creating a geometric

---

[1]Anonymous Institution, Anonymous City, Anonymous Region, Anonymous Country. Correspondence to: Anonymous Author <anon.email@domain.com>.

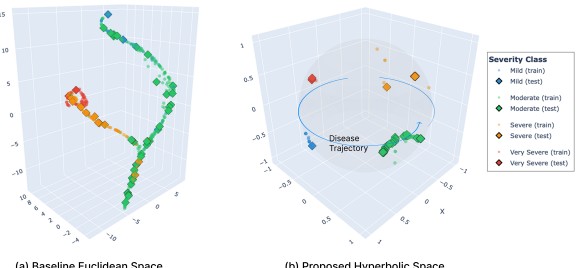

(a) Baseline Euclidean Space    (b) Proposed Hyperbolic Space

*Figure 1.* Visualization of TBI severity embeddings for test samples and their retrieved neighbors in the TBI-MH103 dataset. (a) The Euclidean baseline exhibits crowding and linear overlap between severity classes. (b) Our HICL-TBI approach utilizes the Poincaré Ball to naturally separate hierarchical classes along a distinct disease trajectory, improving discrimination without fine-tuning.

bottleneck for complex, hierarchical data (Bronstein et al., 2017). Traumatic Brain Injury (TBI) prognosis represents a canonical clinical challenge that perfectly illustrates this limitation (Maas et al., 2022; AE et al., 2025; Shickel, 2018). Traditional machine learning often struggles with its noisy, severely imbalanced datasets (Borisov et al., 2022; Emmanuel et al., 2021; Grinsztajn et al., 2022; Zhuang et al., 2020), where clinical severity naturally forms a latent hierarchy progressing from common mild states to divergent severe outliers (Nielson JL, 2015). In flat Euclidean spaces, this hierarchy suffers from severe distortion and the crowding problem (Nickel & Kiela, 2017), causing $L_2$-based retrieval to select semantically irrelevant neighbors and degrade ICL performance in critical high-risk regions.

To overcome this representational limit, we introduce Hyperbolic Retrieval-Augmented In-Context Learning (HICL-TBI). By mapping patient features into a Poincaré Ball—a space with constant negative curvature—we exploit hyperbolic geometry's capacity to naturally model continuous hierarchies and exponentially separate severe clinical outliers (Ganea et al., 2018; Gonzalez-Jimenez et al., 2025). HICL-TBI leverages this geometric alignment to optimize context quality while maintaining a strictly training-free regime. Our contributions are summarized as follows:

- We identify and resolve the representational limits of standard $L_2$-based context retrieval in Tabular Foun-

dation Models. By projecting clinical features into a Poincaré Ball, we empirically demonstrate that hyperbolic geometry naturally disentangles latent hierarchical structures, overcoming the crowding problem inherent in flat Euclidean spaces.

- To preserve mathematical consistency across the inference pipeline, we introduce two novel mechanisms: (1) *Hyperbolic Fréchet Imputation* to reconstruct missing values via the Riemannian center of mass, preventing flat-space distortion; and (2) *Density-Aware Hyperbolic Retrieval* to dynamically adapt the context window size ($K^*$) based on local manifold density, effectively mitigating extreme class imbalance.

- We extensively evaluate HICL-TBI on three highly imbalanced, real-world TBI datasets. Operating in a strictly training-free regime, our framework consistently outperforms strong baselines including tuned Decision Trees, TabNet, and Euclidean TabPFN variants demonstrating the critical value of structured geometric priors in small-data clinical settings.

## 2. Related Work

Tabular Foundation Models like TabPFN (Hollmann et al., 2023) enable training-free In-Context Learning (ICL), offering a paradigm shift over tree-based (Chen & Guestrin, 2016; Grinsztajn et al., 2022) and deep tabular methods (Arik & Pfister, 2021; Gorishniy et al., 2021) that frequently overfit on small clinical datasets (Gravesteijn et al., 2020; Bruschetta et al., 2022; Shwartz-Ziv & Armon, 2022). To scale ICL, retrieval-augmented strategies (e.g., LoCalPFN (Thomas et al., 2024), TabR (Gorishniy et al., 2024)) utilize nearest neighbors. However, their reliance on Euclidean ($L_2$) distance inherently assumes a flat geometry, failing to capture the hierarchical structures of complex biological data. Consequently, critical domains like Traumatic Brain Injury (TBI) prognostication still heavily rely on standard regression baselines (Steyerberg et al., 2008; Perel et al., 2008). Our work addresses this gap by introducing hyperbolic geometry for context retrieval, providing a topologically superior space to unlock Tabular Foundation Models for complex clinical applications.

## 3. Methodology: HICL-TBI

We introduce **HICL-TBI**, a framework designed to overcome the geometric bottleneck of Euclidean retrieval in tabular foundation models by replacing the flat geometry assumption with a negative curvature prior (the Poincaré Ball).

### 3.1. The Euclidean Bottleneck in Tabular FMs

Tabular Prior-Data Fitted Networks (TabPFN) (Hollmann et al., 2023) approximate Bayesian inference for tabular classification via In-Context Learning (ICL). Given a training

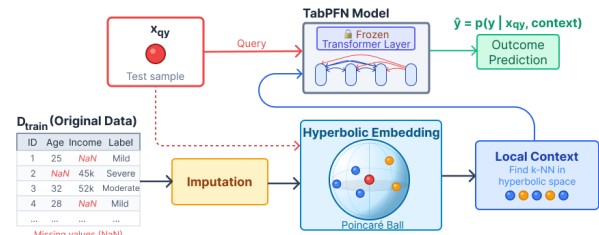

*Figure 2.* The HICL-TBI pipeline: (1) Hyperbolic Fréchet Imputation and scaling, (2) Poincaré projection via the exponential map, (3) Density-aware retrieval based on hyperbolic distance, and (4) In-context prediction via TabPFN.

dataset $\mathcal{D}_{train}$ and a query $x_{query}$, it predicts the posterior in a single forward pass without gradient-based training:

$$P(y_{query}|x_{query}, \mathcal{D}_{train}) = \frac{\exp(f_\theta(x_{query}, \mathcal{D}_{train})\,[y_{query}])}{\sum_{c=1}^{C}\exp(f_\theta(x_{query}, \mathcal{D}_{train})\,[c])} \tag{1}$$

However, TabPFN's efficacy is strictly bound by a context window size ($N_{ctx}$). To scale beyond this, retrieval-augmented strategies like LoCalPFN (Thomas et al., 2024) construct a refined local context $\mathcal{C}_{local}$ composed of $k$ nearest neighbors that minimize a specific distance metric $\delta_{\mathbb{M}}$. This ensures the foundation model conditions its prediction on the most pertinent information:

$$\mathcal{C}_{local} = \underset{\mathcal{S}\subset\mathcal{D}_{train}, |\mathcal{S}|=k}{\arg\min} \sum_{(x_j, y_j)\in\mathcal{S}} \delta_{\mathbb{M}}(x_{query}, x_j) \tag{2}$$

Despite boosting performance, existing methods predominantly rely on Euclidean distance ($\mathbb{M}=\mathbb{R}^d$). As established, complex clinical domains like TBI exhibit a latent hierarchical severity progression. Flat Euclidean geometry lacks the capacity to preserve semantic distances between highly divergent critical cases, leading to the crowding problem (Figure 1). To extract clinically meaningful contexts, HICL-TBI formally shifts the retrieval manifold to a Hyperbolic embedding space ($\mathbb{M}=\mathbb{H}^d$), dynamically adapting to local density.

### 3.2. Modeling TBI Severity in Hyperbolic Space

We hypothesize that TBI severity follows a latent hierarchy, progressing from common mild baseline cases to complex critical outcomes at the boundary of a negatively curved space. We utilize the $d$-dimensional Poincaré Ball model $\mathbb{D}_c^d = \{x \in \mathbb{R}^d : \|x\| < 1/\sqrt{c}\}$ parameterized by curvature $c \geq 0$. Its Riemannian metric tensor is conformal to the Euclidean metric $g^E$ via the conformal factor $\lambda_x = 2/(1 - c\|x\|^2)$.

As $\|x\| \to 1/\sqrt{c}$, $\lambda_x \to \infty$, causing the space's volume to grow exponentially. This provides the geometric capacity to separate severe patients—who often exhibit subtle but criti-

cal feature deviations—effectively mitigating the Euclidean crowding problem (Bronstein et al., 2017).

To map raw, standardized clinical features $x \in \mathbb{R}^d$ onto this manifold, we apply the Riemannian Exponential Map:

$$z = \text{Exp}_0(x) = \tanh(\sqrt{c}\|x\|)\frac{x}{\sqrt{c}\|x\|} \qquad (3)$$

This non-linear squashing maps typical patients linearly near the origin ($\tanh(\|x\|) \approx \|x\|$), while highly divergent severe outliers are compressed towards the boundary. Crucially, due to the exponentially growing conformal factor, these boundary mapped outliers are spread apart geometrically, ensuring a highly discriminative retrieval space.

### 3.3. Hyperbolic Retrieval-Augmented In-Context Learning

As illustrated in Figure 2, the proposed pipeline executes sequentially through three key stages:

**Step 1: Hyperbolic Imputation & Projection.** Standard Euclidean imputation distorts the curved clinical severity space. To preserve geometric consistency, we propose *Hyperbolic Fréchet Imputation*. After a baseline median initialization, features $x^{(0)}$ are projected onto the Poincaré ball via the exponential map (Eq. 3). For each sample $z_i$ containing missing features, we identify its $K$ hyperbolic nearest neighbors $\mathcal{N}_{\mathbb{D}}(z_i)$. Missing values are then iteratively updated by computing the Fréchet mean—the geometric center of mass—of these neighbors:

$$\mu = \arg\min_{m \in \mathbb{D}_c^d} \sum_{z_j \in \mathcal{N}_{\mathbb{D}}(z_i)} \delta_{\mathbb{D}}^2(m, z_j) \qquad (4)$$

This manifold-native centroid prevents the artificial downgrading of severity scores caused by linear interpolation. Once imputed, both the complete training set $\mathcal{D}_{train}$ and query $x_{query}$ reside fully in the Poincaré Ball $\mathbb{D}_c^d$.

**Step 2: Density-Aware Hyperbolic Retrieval.** We compute pairwise Poincaré distances between the query $u$ and training samples $v$:

$$\delta_{\mathbb{D}}(u,v) = \frac{1}{\sqrt{c}}\text{arccosh}\left(1 + 2c\frac{\|u-v\|^2}{(1-c\|u\|^2)(1-c\|v\|^2)}\right) \qquad (5)$$

Given the extreme density variations between dense mild clusters and sparse severe boundaries, a fixed context size $k$ is brittle. We introduce a Parzen-style variable bandwidth mechanism to dynamically determine the context size $k^*$. The query's local scale $\sigma_u$ is defined by its $m_{ref}$-th nearest training point:

$$\sigma_u = \delta_{\mathbb{D}}(u, v_{m_{ref}}) \qquad (6)$$

Using a multiplier $\alpha > 1$, we define an adaptive search radius $r_u = \alpha \cdot \sigma_u$ to retrieve the optimal context set $\mathcal{C}^*$:

$$\mathcal{C}^* = \{v \in \mathcal{D}_{train} \mid \delta_{\mathbb{D}}(u,v) \leq r_u\} \qquad (7)$$

The effective context size $k^* = |\mathcal{C}^*|$ is strictly clipped to $[k_{min}, k_{max}]$. This inherently ensures a compact context in dense regions while automatically expanding the radius to capture a statistically sufficient context in sparse outlier regions.

**Step 3: Asymmetric In-Context Inference.** To maintain distributional alignment with TabPFN's meta-training, we retrieve the original Euclidean features corresponding to the optimal context indices $\mathcal{C}^*$. This context is concatenated with the Euclidean query $x_{query}$ and processed by TabPFN, effectively performing training-free Bayesian inference explicitly conditioned on a geometrically curated patient history.

## 4. Experiments

### 4.1. Datasets

We evaluate HICL-TBI on three real-world clinical datasets encompassing diverse data regimes, missingness patterns, and class imbalances.

(1) **Pilot TBI Cohort** (Bruschetta R, 2022 Mar 16;10(3): An extremely small-scale dataset ($N = 102$, 17 features) predicting 4-class Glasgow Outcome Scale-Extended (GOSE) outcomes, exhibiting severe class imbalance with a 24.5% mortality/vegetative rate. (2) **TBI-MH103:** A newly curated clinical dataset ($N = 504$, 68 features) from 103 Military Hospital, Vietnam. Unlike standard benchmarks, it preserves raw quantitative CT measurements and predicts 4-class severity, heavily challenged by systemic missing values in milder presentations. (3) **ODC-TBI** (Nelson L. D., 2025): A large-scale registry from the TRACK-TBI study ($N = 2545$, 34 acute features). We formulate both an aggregated 4-class and a highly granular 8-class GOSE prediction task (ranging from Death to Upper Good Recovery) to rigorously test hyperbolic disentanglement in high-resolution severity spaces.

### 4.2. Experimental Setup and Results

**Setup.** We evaluate predictive performance using stratified 10-fold cross-validation. Our framework utilizes a frozen TabPFN foundation model ($N_{ens} = 18$) as the inference engine. For density-aware retrieval, we empirically set the reference neighbor =10 and radius multiplier $\alpha$=3.0 across all datasets. To isolate the impact of our geometric prior, we compare HICL-TBI against standard TabPFN, tree-based ensembles, deep tabular models, and crucially, a Euclidean retrieval baseline. We conducted all experiments on one NVIDIA A30 24GB GPU.

**Results.** As summarized in Table 1, HICL-TBI consistently achieves superior performance across diverse data regimes while operating strictly training-free. On the extremely small **pilot TBI dataset**, HICL-TBI achieves the highest overall performance (84.00% accuracy, 92.59% specificity), outperforming standard TabPFN by +1.91%. This confirms

*Table 1.* Comprehensive performance comparison of various models across datasets. All metrics are reported in percentages (%). The best-performing results are highlighted in bold, while the second-best results are underlined.

| Model | Dataset: TBI | | | | Dataset: TBI-MH103 | | | |
|---|---|---|---|---|---|---|---|---|
| | Acc | Prec | Sens | Spec | Acc | Prec | Sens | Spec |
| Logistic Regression | 81.36 | 70.27 | 72.62 | 92.05 | 76.42 | 71.69 | 67.17 | 89.28 |
| SVM | 78.36 | 66.49 | 69.67 | 90.40 | 78.17 | 60.65 | 62.95 | 88.46 |
| Decision Tree | 71.36 | 70.56 | 67.35 | 89.52 | 74.82 | 70.40 | 69.97 | 88.72 |
| Random Forest | 82.09 | 70.78 | 74.25 | 92.27 | 83.55 | 74.53 | 70.56 | 91.14 |
| k-NN | 79.45 | 69.35 | 71.19 | 90.98 | 68.24 | 64.52 | 51.98 | 83.80 |
| TabNet | 73.64 | 65.94 | 65.72 | 8970 | 75.41 | 65.34 | 67.48 | 89.16 |
| TabPFN | 82.09 | 74.35 | 77.37 | 92.50 | 83.35 | 77.71 | 75.19 | 91.60 |
| TabPFN+kNN | 82.09 | 73.99 | 76.12 | 91.96 | 83.72 | 76.45 | 71.61 | 90.48 |
| LoCalPFN | 84.00 | 74.93 | 78.00 | 92.49 | 84.74 | 78.76 | 73.75 | 91.86 |
| **HICL-TBI (ours)** | **84.00** | **75.59** | **78.17** | **92.59** | **86.15** | **79.14** | **78.63** | 91.93 |

| Model | Dataset: ODC-TBI (4-class) | | | | Dataset: ODC-TBI (8-class) | | | |
|---|---|---|---|---|---|---|---|---|
| | Acc | Prec | Sens | Spec | Acc | Prec | Sens | Spec |
| Logistic Regression | 88.62 | 80.82 | 79.10 | 95.98 | 31.61 | 30.04 | 28.23 | 89.66 |
| SVM | 90.40 | 86.14 | 79.82 | 96.52 | 33.05 | 30.11 | 27.75 | 89.83 |
| Decision Tree | 95.22 | 92.93 | 92.70 | 98.35 | 26.50 | 24.80 | 25.38 | 89.02 |
| Random Forest | 95.46 | 93.50 | 92.25 | 98.43 | 32.95 | 33.42 | 29.44 | 89.83 |
| k-NN | 87.91 | 79.99 | 74.30 | 95.57 | 26.85 | 24.49 | 25.26 | 89.10 |
| TabNet | 93.16 | 93.40 | 93.16 | 97.94 | 26.80 | 21.62 | 26.80 | 73.80 |
| TabPFN | **98.59** | **98.08** | **98.56** | **99.59** | 35.48 | 33.54 | 30.50 | 90.15 |
| TabPFN+kNN | 98.34 | 97.11 | 97.66 | 99.42 | 34.89 | 32.43 | 29.79 | 89.96 |
| LoCalPFN | 98.04 | 97.64 | 97.66 | 99.56 | 35.43 | 34.29 | 31.10 | 90.22 |
| **HICL-TBI (ours)** | 98.34 | 97.48 | 97.87 | 99.43 | **36.33** | **34.81** | 31.57 | 90.26 |

that when data is exceedingly scarce, structurally curated hyperbolic contexts are significantly more informative for Bayesian inference than random global subsets.

On the **TBI-MH103 dataset**, HICL-TBI (85.34%) decisively surpasses both standard TabPFN (83.35%) and Euclidean retrieval (83.72%), substantiating that hyperbolic geometry better maps the hierarchical nature of clinical severity. Finally, on the large-scale **ODC-TBI**, while performing comparably on the baseline 4-class task, HICL-TBI achieves the highest accuracy (35.96%) on the highly granular 8-class classification, exceeding both TabPFN (35.48%). These results validate that the Poincaré embedding effectively resolves subtle differences between adjacent severity levels where flat Euclidean metrics succumb to the crowding problem.

### 4.3. Geometric Disentanglement and Theoretical Insights

To elucidate HICL-TBI's performance gains, we analyze its learned representations both visually and theoretically. As shown in Figure 1a, Euclidean embeddings exhibit severe class overlap. Because flat geometry restricts volume growth polynomially, complex phenotypes suffer from the crowding problem. In contrast, the Poincaré projection (Figure 1b) induces a striking radial hierarchy, where *Mild* cases cluster at the origin and *Severe* complications are pushed towards the boundary. This disentanglement is mathematically driven by two intrinsic hyperbolic properties:

**Gromov $\delta$-Hyperbolicity:** Clinical progression resembles a continuous branching tree (from healthy roots to

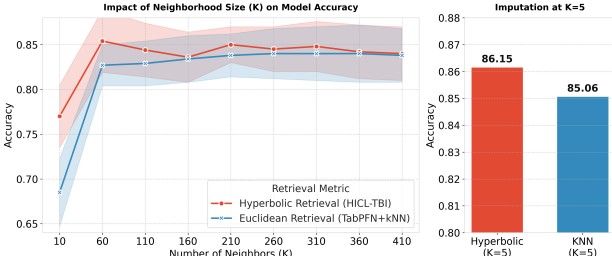

*Figure 3.* **(Left)** Accuracy analysis of retrieval size ($k$) on the TBI-MH103 dataset. **(Right)** Accuracy with the input imputation method using Hyperbolic and Euclidean.

diverse severe trajectories). By Gromov's theorem, tree-like structures can be embedded into hyperbolic space with minimal distortion, enabling TabPFN to retrieve contextually accurate trajectories rather than mixed Euclidean clusters (Nickel & Kiela, 2017).

**Implicit Margin Maximization:** In the Poincaré ball, the distance between two distinct severe phenotypes near the boundary is exponentially amplified compared to their Euclidean separation (Khrulkov et al., 2020). This exponential margin forces heterogeneous severe cases to be pushed infinitely far apart along the boundary, inherently neutralizing the class imbalance problem.

This theoretical margin maximization directly translates to empirical retrieval efficiency. Figure 3 **Left** quantitatively substantiates this: HICL-TBI consistently outperforms the Euclidean baseline across varying context sizes ($k$). The gap is most prominent in low-$k$ regimes, confirming that the exponentially amplified margin reliably isolates semantically accurate neighbors without requiring a large, noise-prone search radius to stabilize. Furthermore, we isolate the impact of our proposed imputation mechanism. As depicted in Figure 3 **Right**, our ablation study on the TBI-MH103 dataset demonstrates that Hyperbolic Fréchet Imputation preserves geometric integrity during data reconstruction, yielding a distinct accuracy gain 86.15% compared to standard Euclidean KNN imputation 85.06% at the same context size (K=5).

## 5. Concluding Remarks

In this paper, we introduce HICL-TBI, a training-free hyperbolic retrieval framework that adapts Tabular Foundation Models (e.g., TabPFN) to the latent hierarchical structures of clinical data. Across three TBI datasets, our density-aware retrieval and hyperbolic imputation effectively mitigate data missingness and severe class imbalance, consistently outperforming Euclidean baselines in small-data and fine-grained tasks. Despite minor computational overhead, HICL-TBI establishes a robust geometric paradigm for In-Context Learning. Future work will explore adaptive curvature and multimodal inputs.

## Impact Statement

This paper presents work whose goal is to advance the field of Machine Learning. There are many potential societal consequences of our work, none which we feel must be specifically highlighted here.

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
