# OpenReview forum: "HICL-TBI: Hyperbolic In-Context Learning for Traumatic Brain Injury Outcome Prediction"
_ICML.cc/2026/Workshop/FMSD — FMSD @ ICML 2026 Poster_

### Official Review · Reviewer_F9nK · 2026-05-18

**Rating:** 5
**Confidence:** 4

**Review:**

## Summary

This paper utilises the hyperbolic geometry in in-context learning for traumatic brain injury prognosis problem.
Unlike classical machine learning problems that TabPFN operates (or its retrieval variant, LoCalPFN), other natural datasets often require non-Euclidean geometry, as in this paper, TBI have similar moderate symptoms, but diverging severe outliers, which is hard to be modeled in Euclidean geometry.
Without retraining the TabPFN network, authors projects back the hyperbolic feature to Euclidean space, utilising hyperbolic geometry to data selection and imputation, and using predictive performance of TabPFN in Euclidean geometry.
Authors test their implementation on three TBI dataset, where authors improve the performance against baseline TabPFN and LoCalPFN, while the improvement is minor.


## Strength

The paper proposes two modifications required to apply LoCalPFN to data with hierarchical structure, specifically, for missing data imputation and data selection.
Both of these algorithms accurately utilises the hyperbolic geometry (mean in hyperbolic geometry, density-aware retrieval), while not distorting the data distribution that would not meet the TabPFN’s prior distribution.
This suggests what is a correct way to use TabPFN on the dataset that would be not on the TabPFN’s prior, we apply techniques in the dataset’s geometry (data batch selection, imputation, etc) then apply the TabPFN on back-transformed dataset.
The suggested methodology shows an accurate approach for one to apply TabPFN (or other tabular foundation models) without leaving the pretrained prior, showing improvement of performance across three out of four dataset.


## Areas for Improvement

Two suggested methods are not uniquely applicable to tabular foundational models, the missing data imputation and mini batch data selection can be applied to classical ML methods as logistic regression or k-NN, which similarly assume Euclidean geometry (max-margin and neighbor selection).
Do these methods only help tabular foundation models, or such models also? Do these methods have specialised benefits on the tabular foundation model?
Also, both missing data imputation and mini batch data selection is a well studied problem in the statistics community, it would be better if authors discuss whether data imputation or mini batch selection is studied under hyperbolic geometry (or more generally, in manifold).

The use of LoCalPFN should be justified in this paper.
TabPFN can handle 1000 training points, which is enough for two datasets (Pilot TBI Cohort, TBI-MH103) and TanPFN2 can handle up to 10000 samples, which can handle all datasets.
Therefore, it is unclear why use of LoCalPFN and hyperbolic retrieval is necessary on these datasets.
A pointer to the large scale dataset that requires hyperbolic geometry would be enough for justification.

## Detailed Comments

In Table 1, there is a typo (TBI dataset, TabNet, Specificity) with missing point. Also, the second-best underline is missing when HICL-TBI is best.

In Section 1, authors argue that TBI dataset is best explained with hyperbolic geometry, as mild / moderate classes are mostly clustered, but severe classes would diverge from.
While introducing hyperbolic representation helped representation geometry as shown in Figure 1, the result is inconsistent with the authors’ claim. Rather, they cluster, but all positioned at the boundary of Poincare disk.
While authors do not explicitly argue how Figure 1 was created, it seems dimensionality reduction method (like PCA) is employed, which might underperform in hyperbolic geometry.
The histogram of the norm of embedding might be suitable to test the authors’ hypothesis.

Figure 3 (Left) is used to show how hyperbolic representation helps low-k regime performance.
However, as the performance gap degrades as k increases, it would be better to decrease grid gap (x-axis), for instance, test the methodology on $k \in \\{10, 20, \ldots\\}.


## Justification of Score

While the authors show a clear method to adapt the tabular foundation models to nonstandard geometry, it is unclear if current methodology is specially designed for tabular foundation models, or is a general methodology for non Euclidean tabular machine learning.

---

### Official Review · Reviewer_vCdx · 2026-05-19
**Hyperbolic representations for improving TabPFN on hierarchical clinical features**

**Rating:** 7
**Confidence:** 4

**Review:**

**Summary**

The authors propose a method to use hyperbolic feature representations to improve the context-size bottleneck of TabPFN and show how the projection of the clinical features on a Poincaré Ball clearly separates the classes following a radial hierarchy, thus improving the classification accuracy on three traumatic brain injury (TBI) outcome prediction tasks. They propose two contributions: (i) Hyperbolic Fréchet Imputation that imputes missing features based by computing the Riemannian center of mass, and (ii) Density-aware hyperbolic retrival, that selects the k-nearest neighbours based on Poincaré distances in the hyperbolic space. Results show consistent improvements over classical baselines and TabPFN variants.

**Strengths**

- This work tries to address the limitations of a flat data manifold (resulting from Euclidean distance-based metrics for selecting features in the context set) by showing the outliers in clinical data (TBI datasets) are mapped to the boundaries of the Poincaré ball in contrast to the crowding and linear overlap exhibited by Euclidean manifolds.

- With tabular foundation models, context set sizes are increasingly becoming the bottleneck, so, having such proof-of-concept results presenting alternative feature projection strategies to enrich the context set is good.

- Concrete demonstration of the crowding of classes when projecting features on the euclidean space in Figure 1 clearly motivates the use of hyperbolic features.

- Despite page-limits, the authors do a good job in giving sufficient background, motivating the problem, and include multiple baselines for comparison.


**Other Comments:**

- From my understand of hyperbolic spaces, the curvature (c) of the Poincaré Ball is critical to ensure a rich enough embedding space; have authors experimented different curvature sizes and can they comment on sensitivity of the model performance to this hyperparameter?

- Related to the above, can the authors specify the values chosen for `c` , k_min, k_max

- Please increase the font size in the final version, the text in the legend is barely visible.